# Anti-prothrombin autoantibodies enriched after infection with SARS-CoV-2 and influenced by strength of antibody response against SARS-CoV-2 proteins

Marc Emmenegger[1], Sreedhar Saseendran Kumar[2☉], Vishalini Emmenegger[2☉], Tomas Malinauskas[3], Thomas Buettner[4], Laura Rose[4], Peter Schierack[5,6], Martin F. Sprinzl[7,8], Clemens J. Sommer[9], Karl J. Lackner[8], Adriano Aguzzi[1‡], Dirk Roggenbuck[4,5,6‡], Katrin B. M. Frauenknecht[1,9‡*]

1 Institute of Neuropathology, University of Zurich, Zurich, Switzerland, 2 Department of Biosystems Science and Engineering, ETH Zürich, Basel, Switzerland, 3 Division of Structural Biology, Wellcome Centre for Human Genetics, University of Oxford, Oxford, United Kingdom, 4 GA Generic Assays GmbH, Dahlewitz, Germany, 5 Institute of Biotechnology, Faculty Environment and Natural Sciences, Brandenburg University of Technology Cottbus-Senftenberg, Senftenberg, Germany, 6 Faculty of Health Sciences Brandenburg, University of Technology Cottbus-Senftenberg, Senftenberg, Germany, 7 Department of Internal Medicine I, University Medical Center of the Johannes Gutenberg-University, Mainz, Germany, 8 Institute of Clinical Chemistry and Laboratory Medicine, University Medical Center of the Johannes Gutenberg-University, Mainz, Germany, 9 Institute of Neuropathology, University Medical Center of the Johannes Gutenberg-University, Mainz, Germany

☉ These authors contributed equally to this work.
‡ These authors are joint senior authors on this work.
* frauenknechtk@gmail.com

## Abstract

Antiphospholipid antibodies (aPL), assumed to cause antiphospholipid syndrome (APS), are notorious for their heterogeneity in targeting phospholipids and phospholipid-binding proteins. The persistent presence of Lupus anticoagulant and/or aPL against cardiolipin and/or β2-glycoprotein I have been shown to be independent risk factors for vascular thrombosis and pregnancy morbidity in APS. aPL production is thought to be triggered by–among other factors–viral infections, though infection-associated aPL have mostly been considered non-pathogenic. Recently, the potential pathogenicity of infection-associated aPL has gained momentum since an increasing number of patients infected with Severe Acute Respiratory Syndrome Coronavirus 2 (SARS-CoV-2) has been described with coagulation abnormalities and hyperinflammation, together with the presence of aPL. Here, we present data from a multicentric, mixed-severity study including three cohorts of individuals who contracted SARS-CoV-2 as well as non-infected blood donors. We simultaneously measured 10 different criteria and non-criteria aPL (IgM and IgG) by using a line immunoassay. Further, IgG antibody response against three SARS-CoV-2 proteins was investigated using tripartite automated blood immunoassay technology. Our analyses revealed that selected non-criteria aPL were enriched concomitant to or after an infection with SARS-CoV-2. Linear mixed-effects models suggest an association of aPL with prothrombin (PT). The strength of the antibody response against SARS-CoV-2 was further influenced by SARS-CoV-2

**Data Availability Statement:** De-identified raw data underlying this study is provided as an archive file in the supplementary information (S1 Data).

**Funding:** We acknowledge funding by a grant of the Innovation Fund (INNOV00096) of the University Hospital Zurich to AA and ME. Institutional core funding by the University of Zurich and the University Hospital of Zurich as well as Driver Grant 2017DRI17 of the Swiss Personalized Health Network and an Advanced Grant of the European Research Council (ERC Prion2020, 670958), and a Distinguished Scientist Award of the Nomis Foundation were granted to AA. The funders had no role in study design, data collection and analysis, decision to publish, or preparation of the manuscript.

**Competing interests:** I have read the journal's policy and the authors of this manuscript have the following competing interests: DR has a management role and is a shareholder of GA Generic Assays GmbH and Medipan GmbH but no financial conflict of interest. Both companies are diagnostic manufacturers. LR and TB are employees of Generic Assays GmbH.

disease severity and sex of the individuals. In conclusion, our study is the first to report an association between disease severity, anti-SARS-CoV-2 immunoreactivity, and aPL against PT in patients with SARS-CoV-2.

## Author summary

An infection with SARS-CoV-2 affects patients differently. While some individuals do not develop any symptoms at all, others become seriously ill, displaying signs of hypercoagulation with involvement of different organ systems. The immune response triggered upon infection may comprise both protective as well as detrimental elements. Among the latter, antibodies targeting a wide range of self-proteins have been evidenced. Our research has focused on autoantibodies targeting phospholipids and phospholipid-binding proteins, held responsible for an increased tendency to thrombosis in an autoimmune disease called antiphospholipid syndrome. Using cohorts from different centres, including patients of mixed disease severities, we found that IgM antibodies targeting prothrombin and β2-glycoprotein I are enriched upon infection. Using a linear mixed-effects model, we further established that anti-prothrombin IgM autoantibodies emerge in proportion to the strength of the antibody response elicited against SARS-CoV-2 proteins, with disease severity and sex as additional contributors. Further research is warranted to investigate the relation between infection with SARS-CoV-2, hypercoagulation, and the presence of autoantibodies directed against prothrombin.

## Introduction

Severe acute respiratory syndrome coronavirus 2 (SARS-CoV-2) was found to elicit a spectrum of autoimmune reactions [1–4], similar to other viral infections [5–7]. Patients with severe SARS-CoV-2 infection, some of whom require mechanical ventilation in specialised hospitals wards, have been shown to be at a high risk of developing thrombotic vessel occlusion [8]. Notably, ischemic events such as stroke have been generally linked with infection, in particular infections affecting the respiratory tract [9]. Along these lines, large-artery ischemic stroke has been identified not only in patients in the current SARS-CoV-2 outbreak [10] but also in 2004 with SARS-CoV-1 [11].

An association of antiphospholipid antibodies (aPL), mainly of the IgA type, and multiple cerebral infarctions has been reported [12], linking SARS-CoV-2 to a systemic autoimmune disease—the antiphospholipid syndrome (APS) [13]. Infection-induced non-criteria aPL [14] could rise in a transient manner and may reflect a non-pathogenic epiphenomenon. Conversely, aPL extracted from SARS-CoV-2 infected patients were reported to induce an accelerated hypercoagulation via activation of neutrophils and release of neutrophil extracellular traps (NETs) that points to a pathogenic role of aPL in SARS-CoV-2 infected individuals [15]. The hypercoagulable state [16] with platelet activation, endothelial dysfunction, increased circulating leukocytes, as well as cytokines and fibrinogen in these patients might be the result of an acquired thrombophilia, as described for patients with APS [17].

To our knowledge, the relationship between criteria and non-criteria aPL and the strength of the antibody response triggered upon infection with SARS-CoV-2 has not been extensively studied. We therefore investigated three cohorts of individuals who contracted SARS-CoV-2 as well as non-infected blood donors in a multi-centre, mixed-severity study. aPL were measured using an established line immunoassay (LIA), including criteria aPL against cardiolipin

(CL) and β2-glycoprotein I (β2) as well as non-criteria aPL detecting phosphatidic acid (PA), phosphatidylcholine (PC), phosphatidylethanolamine (PE), phosphatidylglycerol (PG), phosphatidylinositol (PI), phosphatidylserine (PS), prothrombin (PT), and annexin V (AnV), respectively [18]. Additionally, we used the tripartite automated blood immunoassay (TRABI) technology [19] to investigate anti-SARS-CoV-2 IgG in these cohorts. Overall, our data indicate that PT IgM aPL emerge proportional to the strength of the antibody response elicited against SARS-CoV-2 proteins, with disease severity and sex as additional contributors.

## Methods and materials

### Ethics statement

For this study, we included serum and heparin plasma samples from individuals from Brandenburg/Saxony area, Germany, the University Medical Center Mainz, Mainz, Germany, and the University Hospital Zurich, Zurich, Switzerland. All experiments and analyses involving samples from human donors were conducted with the approval of the local ethics committee (BASEC-Nr. 2020–01731, KEK Zurich; EK2020-16, BTU Cottbus-Senftenberg; reference number 2020-14988-2, ethics committee of the state medical association of Rhineland-Palatinate), in accordance with the provisions of the Declaration of Helsinki and the Good Clinical Practice guidelines of the International Conference on Harmonisation. Written informed consent was obtained from all subjects involved in the study.

### Measurement of autoantibodies against criteria and non-criteria phospholipid and phospholipid-related antigens

Line immunoassays (LIA; GA Generic Assays GmbH, Dahlewitz, Germany) for the detection of criteria and non-criteria antiphospholipid antibodies were used as previously described [20,21]. Briefly, serum and plasma samples were analysed for IgG and IgM autoantibodies against CL, PA, PC, PE, PG, PI, PS, AnV, β2, and PT, according to the manufacturer's recommendations. Diluted samples (1:33, in 10 mM TRIS with 0.1% Tween-20) were transferred onto LIA strips, incubated for 30 min at room temperature (RT) while shaking. A 20 min wash step with 1 ml wash buffer (10 mM TRIS with 0.1% Tween-20) was used to remove unbound or loosely attached unspecific components from the LIA strips. HRP-conjugated anti-human IgM or IgG were incubated for 15 min at RT to bind to autoantibodies. After a subsequent wash step, 500 μl of tetramethylbenzidine (TMB) were added to each LIA strips as a substrate followed by drying the strips for at least 30 min at RT. Optical density (OD) of processed strips were analysed densitometrically using a scanner and the corresponding evaluation software, Dr. Dot Line Analyzer (GA Generic Assays GmbH, Dahlewitz, Germany) with a grayscale calibration card for standardization provided with the kit. The cutoff of 50 OD units was determined by calculating the 99th percentile of 150 apparently healthy individuals as recommended by the international classification criteria for aPL testing and Clinical and Laboratory Standards Institute (CLSI) guideline C28-A3 [18,20]. Matched plasma and serum samples from randomly selected fully anonymised patients with reported aPL-positivity were used to investigate possible differences in aPL pattern detected by LIA. A linear regression was performed and the Pearson correlation coefficient and $R^2$ were calculated, along with Bland-Altman analysis to determine possible bias between serum and plasma measurements.

### High-throughput SARS-CoV-2 serology using TRABI technology

ELISA-based serology was carried out as previously described [19]. In brief, high-binding 1536-well plates (Perkin Elmer, SpectraPlate 1536 HB) were coated on the CertusFlex

dispenser (Fritz Gyger AG) with 3 µL/well 1 µg/mL SARS-CoV-2 spike ectodomain (S), receptor binding domain (RBD), and nucleocapsid protein (NC) in PBS at 37˚C for 1 h, followed by 3 washes with PBS 0.1% Tween-20 (PBS-T) using Biotek El406 and by blocking with 10 µL 5% milk in PBS-T for 1.5 h. Serum or plasma samples were diluted in sample buffer (1% milk in PBS-T) and dispensed using acoustic dispensing technology employing the ECHO 555 (Labcyte). Thereby, we serially diluted the samples in a range between 1:50–1:6,000, at an assay volume of 3 µL/well. After the sample incubation for 2 h at RT, the wells were washed five times with wash buffer and the presence of IgGs directed against above-defined SARS-CoV-2 antigens were detected using an HRP-linked anti-human IgG antibody (Peroxidase AffiniPure Goat Anti-Human IgG, Fcγ Fragment Specific, Jackson, 109-035-098, at 1:4000 dilution in sample buffer), at a volume of 3 µL/well. The samples were then incubated for one hour at RT and subsequently washed three times with PBS-T. Finally, TMB was added using the syringe dispenser on the MultifloFX (BioTek) at the same assay volume as before, plates were incubated for three minutes at RT, and 3 µL/well 0.5 M $H_2SO_4$ were added to stop the chromogenic reaction. The absorbance at 450 nm was measured in a plate reader (Perkin Elmer, EnVision) and the inflection points of the sigmoidal binding curves, i.e. p(EC50) values of the respective sample dilution, were determined using the custom designed fitting algorithm referred to earlier [19].

The assessment of matrix effects of plasma or serum was conducted on patient-matched samples from randomly selected fully anonymised hospital patients (of unknown SARS-CoV-2 vaccination status) using the same pipeline. Plasma/serum dilutions ranging from 1:100–1:72,900 were conducted in technical duplicates and the results were visualised in GraphPad Prism. A linear regression was performed and the Pearson correlation coefficient and $R^2$ were calculated, along with Bland-Altman analysis to determine possible bias between serum and plasma measurements.

## Exploratory data analysis

Pair-wise non-parametric statistical testing was performed to assess differences between controls (non-infected) and SARS-CoV-2 infected individuals. Statistical testing was carried out using MATLAB (MathWorks). Fisher's exact test was performed with two-tailed probability (95% confidence interval, i.e. $\alpha$-level = 0.05) to detect differential distributions of positives/ negatives between two groups. Mann-Whitney/Wilcoxon rank-sum test was performed on groups with significant differences in the Fisher's exact test to assess whether ODs between the two conditions (non-infected/infected) derive from different populations and a Benjamini-Hochberg post-hoc test [22] was applied to account for multiple comparisons. Two-sample Kolmogorov-Smirnov test was used to test for differences in the age distributions between the control and SARS-CoV-2 infected groups. Principal component analysis (PCA) and heatmaps were generated in MATLAB (version 9.10). UMAPs were computed using the umap (https:// CRAN.R-project.org/package=umap) package in R (version 4.03) using default configuration parameters and plotted using ggplot2.

## Development and application of linear fixed-effects and mixed-effects models

We used a linear regression model (fixed-effects) to describe the relationship between a response variable, $y$, (e.g., β2 or PT IgM) and one or more independent variables, $X_i$. The independent variables were a mix of continuously valued covariates (e.g., PC1-SARS-CoV-2-IgG, age, days post onset of symptoms) and categorical factors (sex, severity score, test positivity). A

linear model of the following form was considered:

$$y = \beta_0 + \beta_1 X_1 + \cdots \beta_n X_n + \varepsilon$$

where $\varepsilon$ is the random error.

Least square estimates of the regression coefficients, $\beta_0, \beta_1, \ldots \beta_n$, were computed using the QR decomposition algorithm.

We developed mixed-effects models as an extension of our fixed-effects models. Here, the regression coefficients could vary with respect to one or more grouping variables. In addition to the fixed-effects, these models included random effects associated with individual experimental units drawn at random from a population. Linear mixed-effects models (LMM) of the following form were considered:

$$y = X\beta + Zb + \varepsilon$$

where $y$ is the response variable; $X$ and $Z$ are fixed and random effect design matrices. $\beta$ is a $p$-by-1 fixed-effects vector while $b$ is a $q$-by-1 random effects vector and $p$, $q$ here refer to the number of fixed and random effects respectively in the model. The random effects vector, $b$ and the random error term $\varepsilon$ were assumed to have the following prior distributions:

$$b \sim N(0, \ \sigma^2 D(\theta)) \text{ and } \varepsilon \sim N(0, \ \sigma_\varepsilon^2 I)$$

$D$ is a positive semidefinite matrix parametrized by a variance component vector $\theta$. $I$ is the identity matrix, and $\sigma_\varepsilon^2$ the residual variance. Mixed-effects models were fitted using the maximum likelihood method.

Model development and variable selection was performed using a manual forward step-up procedure. Starting from a constant model, at each step, we explored an alternative model by adding variables–one at a time–either as a fixed or a random effect. The relative quality of the revised model was assessed using the Akaike Information Criterion (AIS), log likelihood and adjusted $R^2$. The revised model was retained only if it passed a likelihood ratio test at an $\alpha$-level of 0.05. The null hypothesis for the test was that the observed response is generated by the simpler model.

## Protein similarity analysis between SARS-CoV-2 Spike and human prothrombin

Amino acid sequences were aligned using EMBOSS Matcher and LALIGN [23]. Structures were analysed and visualized using the PyMOL Molecular Graphics System, Version 2.3.2 by Schrödinger, LLC (https://pymol.org/2/).

## Results

### Multi-centre, mixed-severity study cohort

Non-infected blood donors (controls, n = 20) and samples from individuals who had an RT-qPCR-confirmed SARS-CoV-2 infection (n = 75 samples, from 70 individuals, with 5 repeat samples) were included in our study. Non-infected blood donors had a median age of 47 (interquartile range (IQR): 33–55) years, with 45% of individuals being of female and 55% of male sex (**Table 1** and **S1A** and **S1B Fig**). Individuals who contracted SARS-CoV-2 had a median age of 56 (IQR: 47–70) years and a female-to-male ratio of 41:59.

Twenty-two individuals with a history of SARS-CoV-2 infection were sampled in Brandenburg/Saxony area, Germany, at 59 (IQR:57–87) days post onset (DPO) of symptoms (**Table 2**). They had a severity score of 1, which may include symptoms such as anosmia, fever, fatigue, or

**Table 1. Number of samples (n), median age, and sex distribution of non-infected controls and SARS-CoV-2 infected patients (IQR, interquartile range).**

| Cohort | n | Median age (IQR), years | Sex distribution, ratio |
|---|---|---|---|
| Non-infected controls | 20 | 47 (33–55) | female:male = 45:55 |
| SARS-CoV-2 infected | 75 | 56 (47–70) | female:male = 41:59 |

headache but did not require hospitalization. The cohorts from University Medical Center Mainz (Mainz), Germany (n = 27, of whom 22 were unique patients whereof 5 patients had repeat samples) and University Hospital Zurich (Zurich), Switzerland (n = 26) were cohorts of patients hospitalized due to COVID-19, with severity scores 2 (hospitalization without requiring oxygen supplementation), 3 (hospitalization requiring oxygen supplementation), and 4 (hospitalization with treatment in the intensive care unit (ICU), mostly including ventilation). The sex distribution within each severity score is displayed in **S1C Fig**. The median DPO of symptom for Mainz and Zurich were 13 (IQR: 6–20) and 12 (IQR: 8–15) days, respectively, reflecting earlier time points that still include the acute phase of the infection, unlike for the 22 convalescent individuals who never required hospitalization and were sampled at later time-points after infection. The correlation between severity score and DPO is shown in **S1D Fig**.

## Exploratory analyses indicate association of SARS-CoV-2 infection with autoantibodies against β2-glycoprotein I, and prothrombin

We used an extended IgG and IgM panel of the LIA [20,21] to measure autoantibodies against criteria and non-criteria aPL, including CL, PA, PC, PE, PG, PI, PS, AnV, β2, and PT, in heparin plasma and serum samples of SARS-CoV-2 infected individuals and non-infected controls. For the individuals pertaining to the Zurich cohort, the panel could be applied only to the measurement of IgM aPL due to insufficient sample volume. We assessed the consistency of LIA using matched serum and plasma samples of one patient with reported aPL positivity as well as of a serum sample of one patient with high aPL titer spiked at 1:2 and 1:5 into aPL negative serum and plasma. We found the same pattern of IgM and IgG aPL-positivity in both matched serum and plasma samples as well as in spiked serum and plasma samples, with a Pearson correlation coefficient of 0.9974 (95% confidence intervals: 09611–1.034) and $R^2$ of 0.9813 (**S2A Fig**). Bland-Altman analysis indicated a bias of -2.717 (95% confidence interval: -17.53–12.1) (**S2B Fig**), suggesting that both serum as well as plasma samples behave in a comparable manner without introducing bias in the subsequent analyses.

We aimed to gain insights into the reactivity profile by exploring all data to detect potential patterns. Following this step, models that explicitly account for heterogeneity in all features were included in the analysis, to rigorously follow up on initial findings. We first generated a heatmap of the respective IgM (**Fig 1A**) and IgG aPL profiles (**Fig 1B**). IgG aPL were typically absent and only rarely close to or above the threshold of OD ≥ 50, a cutoff determined previously by calculating the 99th percentile of 150 apparently healthy individuals [18,20], in both

**Table 2. Number of samples (n), severity group, and median day post onset (DPO) of symptom of patients who contracted SARS-CoV-2, belonging to three cohorts (IQR, interquartile range).**

| Cohort | n | Severity group | Median DPO (IQR), days | Sample type |
|---|---|---|---|---|
| Brandenburg/Saxony | 22 | 1 | 59 (57–87) | Serum |
| Mainz | 27 | 2–4 | 13 (6–20) | Serum |
| Zurich | 26 | 2–4 | 12 (8–15) | Heparin plasma |

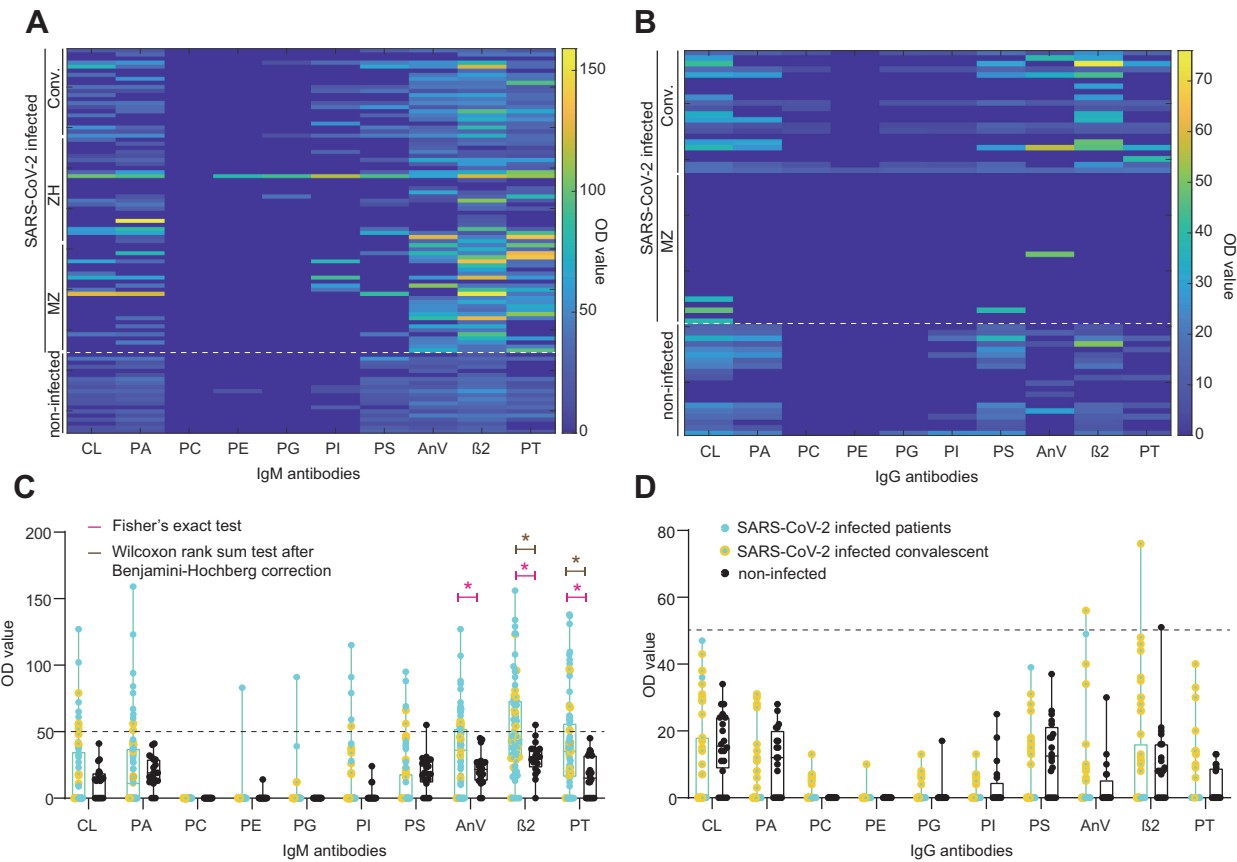

**Fig 1. Heatmaps and boxplots for IgM and IgG aPL. A**. and **B**. Colour-coded representation of IgM (A) and IgG (B) aPL. Higher OD values are evident for AnV, β2, and PT IgM aPL. **C**. and **D**. Boxplot rep-resentation of IgM (C) and IgG (D) aPL. Dotted line: OD value 50, a cutoff determined previously by calculating the 99[th] percentile of 150 apparently healthy individuals (Roggenbuck *et al*., 2016; Nalli *et al*., 2018). Pink star: statistically significant distributional differences between non-infected controls and SARS-CoV-2 infected individuals according to Fisher's exact test. Brown star: statistically significant distributional differences between non-infected controls and SARS-CoV-2 infected individuals according to Wilcoxon rank sum test after Benjamini-Hochberg correction. Black dots: Non-infected controls. Turquoise dots: SARS-CoV-2 infected hospital patients. Yellow dots: SARS-CoV-2 infected convalescent individuals.

non-infected controls and SARS-CoV-2 infected individuals. Although below threshold, the convalescent individuals characterised by a larger DPO had higher IgG titres, as is visible in **Fig 1B**. However, IgM aPL titres were generally higher than the corresponding IgGs. PC, PE, and PG IgM aPL did not show any reactivity (median OD 0, for all the three antibodies), with none (PC IgM) or one (PE and PG IgM) individual having values above threshold. Conversely, IgM aPL against CL, PA, PI, PS, AnV, β2, and PT showed a heterogeneous pattern with titres in both the non-infected controls as well as the SARS-CoV-2 infected individuals, including acutely infected patients as well as convalescent individuals.

We then investigated whether the overall IgM or IgG aPL profiles were distinct between the SARS-CoV-2 infected individuals and the non-infected controls. We used Uniform Manifold Approximation and Projection (UMAP) to reduce the dimensionality of the dataset while preserving the maximum variability, accounting for potential nonlinear relationships. Neither IgM (**S2C Fig**) nor IgG profiles (**S2D Fig**) displayed clear clusters, suggesting that possible differences between non-infected controls and SARS-CoV-2 infected individuals could not be explained in the feature space and require a more granular analysis. On the positive side, the

absence of distinct clusters suggests that SARS-CoV-2 infection does not lead to a global and broad dysregulation of aPL.

We subsequently categorised IgM and IgG aPL data according to an OD threshold of 50, with values $\geq$ 50 considered positive, and values < 50 negative. Additionally, we split the SARS-CoV-2 infected fraction into patients (turquoise dots) and convalescent individuals (yellow dots) for graphical representation. Using Fisher's exact test, we found significant distributional differences between non-infected controls and SARS-CoV-2 infected individuals for AnV IgM (p-value = 0.0026), β2 IgM (p-value = 0.0012), and PT IgM (p-value = 0.0052) (**Fig 1C**) but for none of the IgGs (**Fig 1D**). To further increase the stringency of our analysis, we subjected AnV, β2, and PT IgM to the Wilcoxon rank sum test, followed by the Benjamini-Hochberg correction for multiple comparisons. Here, we aimed to infer whether the probability of an OD value randomly drawn from the control group being greater than one drawn from the infected population was higher than chance level ($\alpha$-level = 0.05), considering the entire distribution of data without setting a cutoff. We identified significant distributional changes for β2 (p-value = 0.005), and PT (p-value = 0.005) IgM but not for AnV IgM (p-value = 0.13, i.e., non-significant). Thus, only β2 and PT IgM displayed statistical significance when applying both criteria, indicating robust SARS-CoV-2 associated changes.

We further evaluated these changes in the light of potentially confounding factors when performing pair-wise testing. We first performed a stratified analysis of associations between the response variable and each of the potential confounding factors: sex and age. To confirm our findings, we used multivariate regression and a percent change-in-estimate criterion. A 10% change or more is commonly used as an indicator of a confounding effect [24,25].

We then explored the influence of sex by stratifying the dependent (β2 IgM and PT IgM OD values) and independent variables (infection status or test positivity) by sex. No trends were found between male and female OD values in each group. Males and females were also similarly distributed among non-infected controls and SARS-CoV-2 infected individuals (**S1B Fig**). These observations suggest that sex is unlikely to be a confounder. We confirmed this using multivariate regression. Briefly, an ordinary least square regression model was fitted between OD values and infection status. The estimate of the regression coefficient associated with SARS-CoV-2 positivity was highly significant (p-value = 0.008 for PT IgM and 0.001 for β2 IgM). We then added sex as an additional independent variable to the regression equation while observing the change in estimate of the coefficient associated with SARS-CoV-2 positivity. The percent change in the coefficient estimate was less than 4% for both β2 and PT IgM. Hence, we ruled out sex as a confounding factor in our analysis.

We next assessed the confounding effect of age in the comparisons between non-infected and SARS-CoV-2-infected groups (see **S1A Fig**). Although the age distribution varied between the control and SARS-CoV-2 infected groups (p-value = 0.009, two-sample Kolmogorov-Smirnov test), we found no significant correlations between age and (β2 or PT IgM) OD values, either overall, or within the non-infected and SARS-CoV-2-infected groups. Moreover, we discovered that the estimate of the coefficient associated with positivity remained stable when age was added as an additional covariate. The percent change in estimate was less than 5% for both β2 and PT IgM. Hence, age was also ruled out as a confounder in our analysis.

In sum, while IgG titres were almost entirely absent in acutely infected patients, they were tendentially increased, even if low and below threshold, in convalescent individuals with a median DPO of 59 days (**Fig 1D**). As our findings resulting from our exploratory analysis suggested that β2 as well as PT IgM values were reliably upregulated as a function of infection with SARS-CoV-2 (see test statistics used in **Table 3** and **Fig 1C**), we decided to focus on β2 as well as PT IgM in further analyses.

**Table 3. Overview of antiphospholipid antibodies (aPL) and pair-wise statistical testing (AnV, annexin 5; β2, β2-glycoprotein I, IQR, interquartile range; CL, OD, optical density; PA, phosphatidic acid; PC, phosphatidylcholine; PE, phosphatidylethanolamine; PG, phosphatidylglycerol; PI, phosphatidylinositol; PS, phosphatidylserine; PT, prothrombin).**

| aPL | | Median (IQR), OD | | % of individuals above nominal cutoff (OD ≥ 50) | | p-value (Fisher's exact test) | p-value (Wilcoxon rank sum test, Benjamini-Hochberg) |
|---|---|---|---|---|---|---|---|
| | | Non-infected controls | SARS-CoV-2 infected | Non-infected controls | SARS-CoV-2 infected | | |
| CL | IgG | 15.5 (9.5,24) | 0 (0, 18) | 0 | 0 | ns | - |
| | IgM | 14 (0, 18.5) | 0 (0, 34) | 0 | 12 | ns | - |
| PA | IgG | 12(0, 19) | 0 (0, 0) | 0 | 0 | ns | - |
| | IgM | 19 (12.5, 29) | 11 (0, 37) | 0 | 14.67 | ns | - |
| PC | IgG | 0 (0, 0) | 0 (0, 0) | 0 | 0 | ns | - |
| | IgM | 0 (0, 0) | 0 (0, 0) | 0 | 0 | ns | - |
| PE | IgG | 0 (0, 0) | 0 (0, 0) | 0 | 0 | ns | - |
| | IgM | 0 (0, 0) | 0 (0, 0) | 0 | 1.33 | ns | - |
| PG | IgG | 0 (0, 0) | 0 (0, 0) | 0 | 0 | ns | - |
| | IgM | 0 (0, 0) | 0 (0, 0) | 0 | 1.33 | ns | - |
| PI | IgG | 0 (0, 3) | 0 (0, 0) | 0 | 0 | ns | - |
| | IgM | 0 (0, 0) | 0 (0, 0) | 0 | 6.67 | ns | - |
| PS | IgG | 12.5 (0, 20.5) | 0 (0, 0) | 0 | 0 | ns | - |
| | IgM | 18.5 (12.5, 28) | 0 (0, 17.5) | 5 | 6.67 | ns | - |
| AnV | IgG | 0 (0, 3.5) | 0 (0, 0) | 0 | 2.04 | ns | - |
| | IgM | 21.5 (13, 27) | 36 (0, 52) | 0 | 29.33 | 0.0026 | ns |
| β2 | IgG | 8 (0, 16) | 0 (0, 14.5) | 5 | 2.04 | ns | - |
| | IgM | 31 (24, 37) | 45 (30.25, 72.5) | 5 | 42.67 | 0.0012 | 0.005 |
| PT | IgG | 0 (0, 8.5) | 0 (0, 0) | 0 | 0 | ns | - |
| | IgM | 15 (0, 32) | 35 (16, 54.75) | 0 | 28 | 0.0052 | 0.005 |

## Titre determination of antibodies directed against three SARS-CoV-2 proteins using the TRABI technology

The clinical picture of SARS-CoV-2 infection is diverse [26,27], the manifestation of first symptoms is highly individual, and its documentation dependent on the governance of the clinical department or the clinician. We have therefore aimed to obtain additional data by determining the respective SARS-CoV-2 antibody titres using the TRABI technology [19], to better characterise the immune profile. First, we analysed patient-matched serum and plasma samples to assess whether its matrix would lead to discordant results. The binding curves (S3A Fig) as well as the resulting p(EC50) values, i.e., antibody titres, were highly congruent (S3B Fig), with a Pearson correlation coefficient of 0.9942 (95% confidence interval: 0.9865–0.9975) and $R^2$ of 0.9885. The same data was plotted as difference in p(EC50) of serum and plasma and their average (Bland-Altman plot). The calculated bias was extremely low (0.01976, standard deviation: 0.1361) and all data points but two, at very low p(EC50) values, resided within the 95% confidence intervals (-0.2471 to 0.2866), displayed in S3C Fig. This suggested that IgG titres derived from both sample types are comparable. Since the SARS-CoV-2 antibody titres were already available for the cohort from Zurich (published in [19]), we measured IgG antibodies against the SARS-CoV-2 spike ectodomain (S), its receptor-binding domain (RBD), and the nucleocapsid protein (NC), in all additional individuals in this study, including the non-infected controls. To this end, eight dilutions (range: 1:50–1:6,000) per sample and antigen were conducted using acoustic dispensing technology and the values were fitted with a logistic regression whereby the p(EC50) was derived, as previously shown [19]. We thus

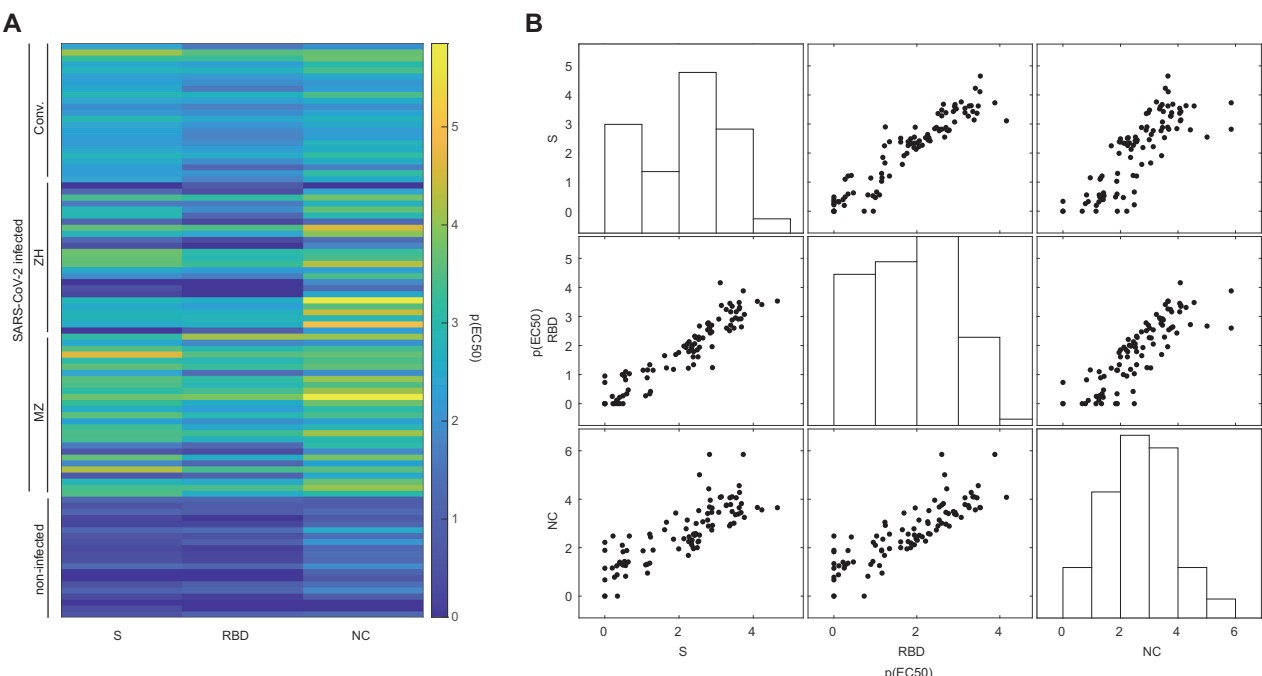

**Fig 2. Antibodies against SARS-CoV-2 proteins using a tripartite autoimmune blood immunoassay. A**. Colour-coded representation of the IgG antibody reactivity profile of non-infected controls and SARS-CoV-2 infected individuals for the SARS-CoV-2 spike protein (S), its receptor binding domain (RBD), and the nucleocapsid protein (NC). The p(EC50) value of the respective sample dilution reflects the inflection point of the logistic regression. **B**. Multicollinearity plot to display the individual reactivity profile of distinct anti-SARS-CoV-2 IgG antibodies. Antibodies against S, RBD, and NC are approximately linear against each other, indicating that information of one is predictive for the other. Bars represent the respective distributions of p(EC50) values obtained for S, RBD, or NC.

obtained matched SARS-CoV-2 IgG titres and LIA results, for all samples and all cohorts. We then visualised the respective titres in a heatmap (**Fig 2A**). Visibly, some of the individuals infected with SARS-CoV-2 displayed titre values in the range of the non-infected controls, most likely because IgG seroconversion had not yet occurred at the time point of sampling. However, the overall separation between non-infected controls and individuals who contracted SARS-CoV-2 was obvious when applying UMAP (**S3D Fig**).

We next illustrated the apparent multicollinearity of the IgG antibody response among S, RBD, and NC (**Fig 2B**) and applied principal component analysis (PCA), to obtain linear combinations. The first principal component (PC), named PC1-SARS-CoV-2-IgG, accounted for 90.9% (second PC: 7.2%, third PC: 1.9%) of the variability contained within the p(EC50) titres and could therefore be reasonably employed to represent the IgG response against SARS-CoV-2 proteins as a composite metric, explaining most of the variability, in subsequent analyses.

## Linear mixed-effects model corroborates the relationship between SARS-CoV-2 infection and aPL against prothrombin and is associated with strength of the antibody response, disease severity, and sex

Autoantibody responses concomitant to or following a viral infection could be driven by many parameters, including the strength of the specific immune response to components of the viral pathogen. Some of these features are supposedly independent, others inter-dependent, and the hierarchy of the contributors is unclear, suggesting mixed effects. Aiming to further investigate

the relationship between infection to SARS-CoV-2 and IgM aPL against β2 and PT, we there-fore chose to utilise a linear mixed-effects model (LMM). Such models allow us to analyse het-erogeneous hierarchical data i.e., data that could vary, or may be grouped by multiple factors with observations across groups exhibiting potentially complex association structures [28]. By accounting for correlations among observed variables LMMs enable us to study the variation of a dependent variable in terms of components corresponding to each level of the hierarchy [29]. Though LMMs make assumptions on the distributions of residuals and random effects, they are remarkably robust to violations of these assumptions [30]. For these reasons, they find broad application in life science, psychology, and medicine (see e.g. [31–34]).

Essentially, the main assumption is that a dependent variable is linearly related to fixed or random factors, with the fixed effects modelling the mean of the dependent variable and the random effects its variability within and between each grouping variable. Along these lines, while so far, our data indicated that an infection with SARS-CoV-2 could influence IgG or IgM aPL levels, we made no assumptions over which additional parameters would be influen-tial. We have therefore chosen a stepwise exploratory approach.

First, using all values available and without segregating non-infected controls and acutely infected and convalescent SARS-CoV-2 infected individuals, we inspected β2 and PT IgM aPL levels as a function of PC1-SARS-CoV-2-IgG levels, looking at sex, disease severity, DPO, and age (**Fig 3A**). Visually, data seemed indicative of a potential effect of sex on PT IgM levels, however, only in dependency of PC1-SARS-CoV-2-IgG levels. Additionally, severity but not age might be partially predictive of β2 and PT IgM levels (**S4A Fig**), while DPO did not display a consistent linear relation and appeared to peak at approximately 20–30 DPO for PT IgM aPL (**S4B Fig**). As a point of caution, severity is not equally distributed along the DPO in our data-set (**S1D Fig**), which needs to be considered when interpreting IgM levels as a function of DPO; we take care of this complexity using an LMM. Additionally, the distributions of β2 and PT IgM were skewed, with an enrichment at OD 0 as well as at ca. OD 40, leading to the appearance of a quasi-binomial distribution (**Fig 3A**). To account for this, individuals with OD < 5 for a given antigen were removed from the subsequent regression analysis. Thus, we aimed to specifically investigate the most important factors regulating the presence, and not the absence, of IgM aPL against β2 and PT. Generally, we first fitted an ordinary least square regression model, then added variables as fixed and as mixed effects and assessed general model parameters (Akaike Information Criterion (AIC), log-likelihood, adjusted $R^2$, and the p-value of the likelihood ratio) and whether the slopes or intercepts improved. While we observed that the fits for both β2 and PT improved when including PC1-SARS-CoV-2-IgG for prediction, none of the additional variables added as a fixed- or mixed-effect were informative in predicting the best fit for β2 IgM. β2 IgM values, in general, were not found to be well explained by a linear model, fixed or mixed. Even if the inclusion of DPO seemed to be infor-mative according to the likelihood ratio, it caused a change in the estimate of PC1-SARS-CoV-2-IgG by 55% and worsened the adjusted $R^2$. Thus, DPO should be interpreted as a con-founder for β2 IgM, in this context. Conversely, the inclusion of sex proved informative on the intercept as well as the slope for PT IgM, and the information contained in DPO further refined the model, in addition to PC1-SARS-CoV-2-IgG (see **S1 Table**). The best fits for both β2 as well as PT IgM are plotted in **Fig 3B**. Although the mixed-effect model already accounted for differences between the acute and the convalescent groups in terms of DPO, we addition-ally removed the convalescent individuals from the analysis to subject the finding to scrutiny. The best fit for PT IgM included PC1-SARS-CoV-2-IgG and sex (see **S2 Table**), as above, and is plotted in **Fig 3C**. Thus, the above identified relation is consistent after the exclusion of con-valescent individuals.

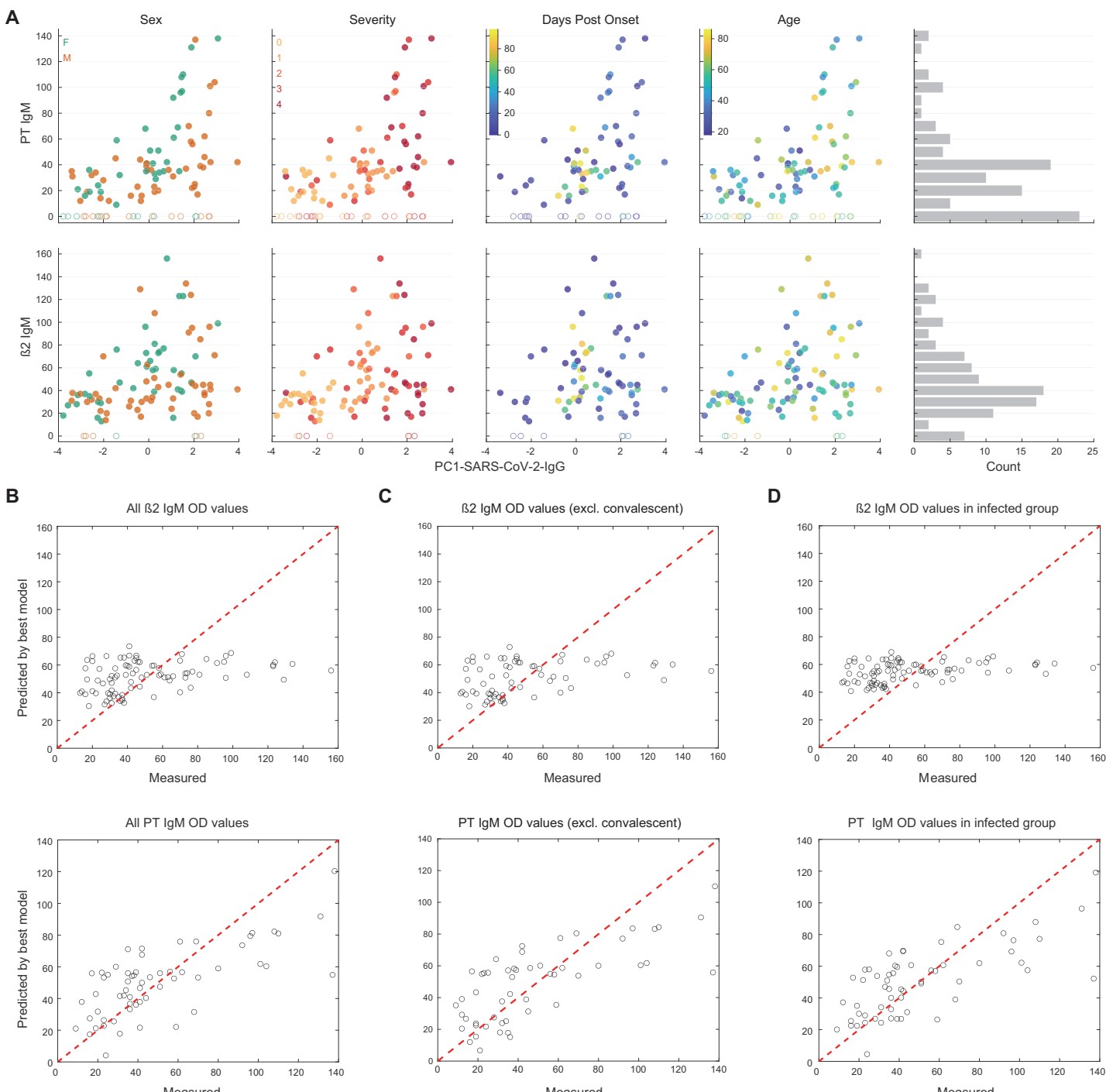

**Fig 3. Exploratory multi-parametric data analysis and best-fit model for β2 and PT IgM. A**. Data exploration to inspect potential relationships between OD values for PT or β2 IgM with PC1-SARS-CoV-2-IgG, sex (f for female, m for male), disease severity (scores 0–4), days post onset of first disease manifestation, or age (in years). A histogram of OD values was included to display the relative frequencies. After a peak at ca. OD value 40, a second peak at 0 emerges for both PT as well as for β2 IgM. **B**. The fitted vs. observed β2 and PT IgM values. While for β2 IgM, only PC1-SARS-CoV-2-IgG is informative, PC1-SARS-CoV-2-IgG, disease severity, and sex, all contribute to the accurate prediction of OD values of PT IgM. **C**. Same as (B) in the absence of convalescent individuals. **D**. Same as (B) in the absence of non-infected controls.

Finally, we investigated the relationship between PC1-SARS-CoV-2-IgG and β2 or PT IgM levels strictly in the fraction of individuals who contracted SARS-CoV-2, using a similar approach to the one described above (see **S3 Table**). This is an important addition as we needed to ensure that the weak antibody score, PC1-SARS-CoV-2-IgG, characteristic for non-infected individuals (see **Figs 2A** and **3A**) is not biasing the analyses of those individuals who contracted SARS-CoV-2. While β2 IgM levels did not display robust improvements upon inclusion of age, sex, DPO, disease severity (in line with the model that includes the non-infected controls), or PC1-SARS-CoV-2-IgG (opposed to the model that includes the non-infected controls), the best fit model indicated that the addition of the composite metric (PC1-SARS-CoV-2-IgG), the severity score, and sex were informative to predict PT IgM levels. The best models for β2 and PT IgM, in the absence of non-infected controls, are shown in **Fig 3D**. We thus conclude that PT IgM aPL levels are mostly associated with the strength of the antibody response elicited against SARS-CoV-2 proteins tested here and are further influenced by disease severity and sex.

## Anti-PT autoimmunity is likely not mediated by cross-reactive IgM antibodies to SARS-CoV-2 Spike protein

Given aPT antibodies are enriched after SARS-CoV-2 infection, we asked whether SARS-CoV-2 Spike could have elicited the production of cross-reactive antibodies of the IgM iso-types, recognising both Spike and PT. To address this question, we identified similarities between Spike and PT amino acid sequences using two methods (EMBOSS Matcher and LALIGN) [23]. Furthermore, to assess whether shared sequences could be recognized by anti-bodies, we mapped identified sequences onto PT and Spike structures [35,36]. Although two proteins differ in their domain architecture, structural analyses revealed four regions of Spike that are similar to PT (sequence identity 27–35% for 21–76 aligned residues; **Fig 4A**). All four regions encompass residues that are exposed on protein surfaces and, thus, could be recog-nised by antibodies (**Fig 4B** and **4C**). However, region 1 contains an N-linked glycosylation site that could interfere with the Spike/PT–antibody interactions. Region 2 contains the PRTF motif shared by two proteins, but they adopt rather different conformations (**Fig 4D**). Regions 3 and 4 lack continuous stretches of identical residues that are longer than 2–3 amino acids. While detailed analyses of the degree of convergence of complementarity-determining regions (CDRs) between antibodies against SARS-CoV-2 Spike and human PT may be informative, prothrombin and Spike share rather limited similarity and, thus, seem less likely to be recog-nised by the same antibody.

## Discussion

Here, we studied patients who contracted SARS-CoV-2 for the occurrence of aPL, in three cohorts originating from three different centers, with mixed disease severity scores. In general, we have chosen a largely data-driven approach. While our main hypothesis was that an infec-tion with SARS-CoV-2 may enhance IgG and/or IgM aPL levels, we had no prior assumption regarding parameters that could potentially be influential. We have therefore collected avail-able parameters and, following a first assessment of which aPL are altered and, thus, worth pursuing, analysed them in a stepwise exploratory manner. Moreover, we established their relationship using linear mixed-effects models, aiming to identify parameters and their combi-nations that best predict aPL levels while accounting for redundant and heterogeneous fea-tures. Lastly, we speculated that cross-reactive IgM antibodies against the SARS-CoV-2 Spike protein could trigger autoimmunity against prothrombin.

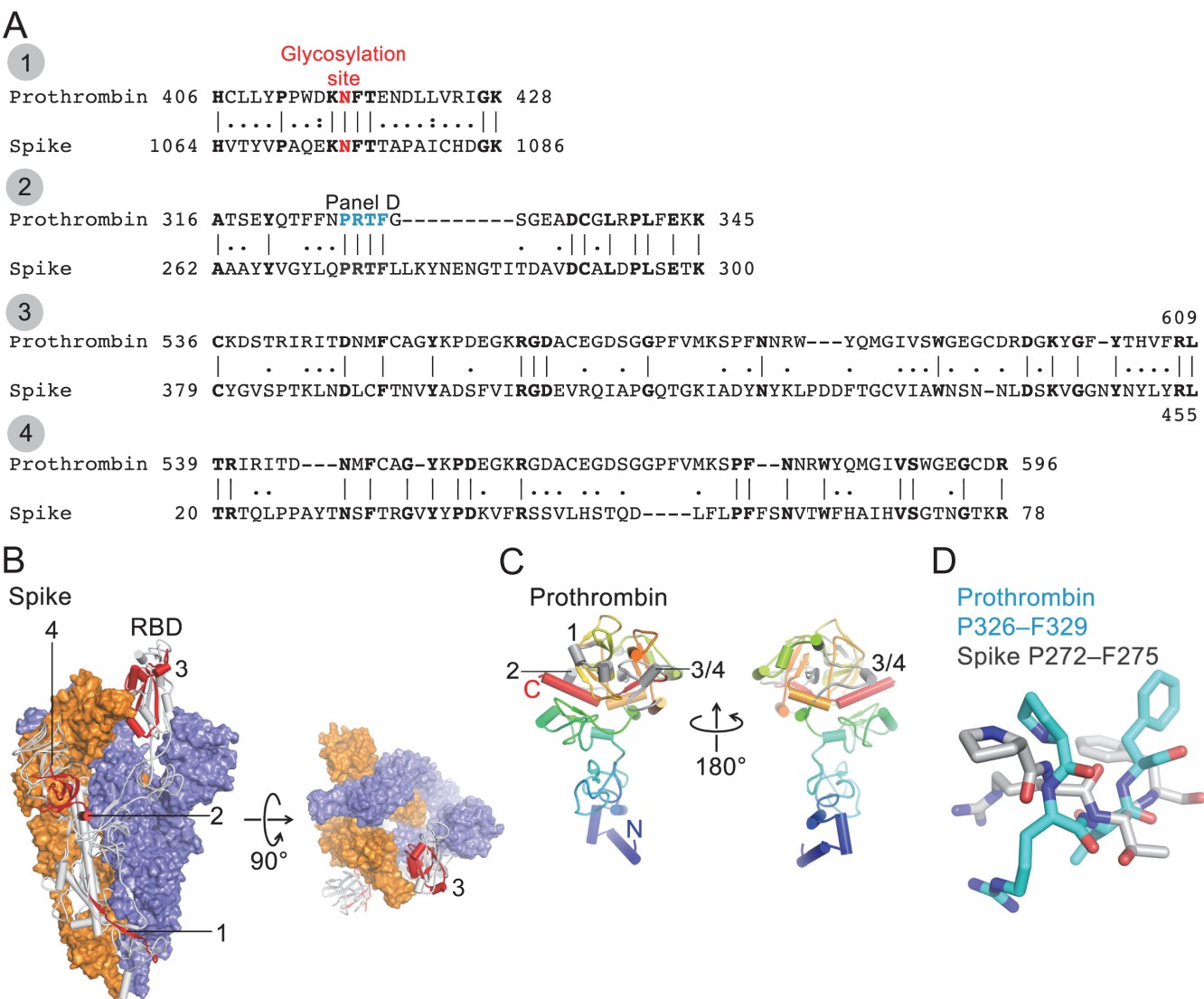

**Fig 4. Sequence analysis of SARS-CoV-2 Spike and human prothrombin. A**. Sequence alignments of SARS-CoV-2 Spike (numbering corresponds to UniProt ID P0DTC2) and human prothrombin (P00734). **B**. Cryo-EM structure of trimeric Spike (Protein Data Bank ID 6Z97 [36]). Two protomers are shown in surface representation (blue and orange) and one as a grey cartoon with 4 peptide regions shown in red and indicated. Region 3 is located in the receptor-binding domain (RBD). Glycans are not shown for clarity. **C**. Crystal structure of prothrombin (PDB ID 5EDM, [35]) shown as a cartoon (N-terminus, blue; C-terminus, red). 4 peptide regions are shown in grey and indicated. **D**. Structural superposition of the PRTF motif from Spike and prothrombin.

We first measured IgG and IgM aPL against criteria (CL and β2) and non-criteria antigens (PA, PC, PE, PG, PI, PS, PT, AnV) and then supplemented our dataset with detailed information on the antibody status of all participants by measuring the presence of IgG directed against SARS-CoV-2 S, RBD, and the NC protein. Our cohorts comprised patients presenting with uncomplicated, mild, moderate and severe disease courses of COVID-19 as well as healthy blood donors who had not contracted SARS-CoV-2. To characterise these patients, we availed of features such as aPL levels, SARS-CoV-2 antibody titres, disease severity, basic demographic information (sex and age), and DPO of sample.

Prior knowledge suggests the presence of a plethora of aPL, including LA, in COVID-19 [13]. Moreover, severe disease courses, including COVID-19-associated coagulopathy (CAC)

are reminiscent of so-called catastrophic APS (CAPS), which features venous and/or arterial vascular thrombosis as well as pulmonary and heart damage with endothelial injury and microthrombosis [37–41]. CAPS is an utterly devastating disease with around 30% mortality [42], in which patients usually develop multiple organ damage over a short period of time [43,44]. CAPS appears to be linked to infections in the first place in nearly half of the patients [42]. Uncontrolled complement activation may further contribute to an unfavourable disease course [42]. However, previous studies on COVID-19-associated aPL (mainly investigating ß2, CL IgM and/or IgG and LA) came to different conclusions regarding their significance in modulating thrombosis. Some studies found no direct relationship of aPL-positivity with thrombotic events [8,45–49], which could be due to different epitope specificity, i.e. different aPL subtypes [50,51] compared to APS. Conversely, other studies found an association of elevated aPL (e.g. LA) with thrombotic events and/or critical illness [52–54].

In our study, we pursued two main objectives. We (1) aimed to provide further evidence, or its absence, for the occurrence of aPL as a result of infection with SARS-CoV-2. If aPL can be linked to infection, we (2) aimed to identify potential correlates. Indeed, we found that, globally, IgM or IgG levels are increased upon infection with SARS-CoV-2, with 66% of individuals having aPL against ≥1 antigen (non-infected controls: 15%), 40% against ≥2 antigens (non-infected controls: 0%), and 21.3% against ≥3 antigens (non-infected controls: 0%), using a threshold of OD ≥ 50. Thus, the prevalence of aPL was higher in our study than previously reported [13,15,50,51,55], despite not including LA in the measurements, and in spite of omitting IgA aPL. Importantly, we detected significant distributional changes between non-infected controls and SARS-CoV-2-infected individuals for IgM aPL against AnV, β2, and PT using Fisher's exact test (p-values < 0.01), and for β2, and PT using Wilcoxon rank sum test (p-values < 0.01 after Benjamini-Hochberg correction). Hence, in our study, we found an association of IgM, and not of IgG, aPL with SARS-CoV-2 infection. The absence of association with IgG aPL is likely caused by our cohort, consisting of (1) mostly acutely infected hospital patients and (2) few convalescent individuals characterised by a mild disease course. As disease severity is a predictor of PT IgM aPL titres and since IgG seroconversion is a time-sensitive process, it is likely that a follow-up of severely diseased hospital patients at later time points after infection may evidence significantly increased levels of IgG aPL. However, our study, while attempting to be as inclusive and comprehensive as possible, only allows to capture information from a limited timeframe, from hospital patients with moderate-to-severe disease courses due to infection with SARS-CoV-2 and from convalescent individuals who had mild symptoms of disease. As such, following an exploratory step, we have decided to focus on β2 and PT IgM aPL. Yet, the immunoepidemiology of IgG aPL at later timepoints after infection with SARS-CoV-2 warrants further investigation, in individuals with both mild as well as more severe disease courses.

Polyreactive circulating IgM antibodies can bind to membrane phospholipids [56] and are supposed to have a protective function [57]. In contrast, such antibodies may not only clear the system from damaged cells but may also drive subsequent cell damage by additional complement activation [58]. In the context of infection with SARS-CoV-2, increased cell death/apoptosis has been described [59]. Specifically, phospholipid-rich pulmonary surfactant leakage [60] following SARS-CoV-2-induced pulmonary cell necrosis may further trigger the rise of aPL. AnV, β2, and PT IgM aPL are enriched after SARS-CoV-2 infection and may last longer than three months, at least in a subset of COVID-19 patients [61]. AnV aPL have been linked to a (pro)thrombotic state in several diseases including sickle cell disease [62] and APS [63]. Additionally, AnV IgG or IgM were associated with higher occurrence of pulmonary arterial hypertension and were detectable throughout a 2-year follow-up in patients with systemic sclerosis [64]. β2 and PT aPL were reported to cause LA (1) via direct interaction of β2

aPL with FV and activation by FXa and (2) via PT aPL competition with FXa for PL binding sites [65], actions which may account for the higher prevalence of LA described in patients with COVID-19 [13]. However, a very recent study indicated LA as a transient phenomenon during SARS-CoV-2 infection [61]. We then decided to focus on β2 and PT IgM aPL and asked what features were predictive of their occurrence. To this end, we started with an ordinary least square regression model, then added variables as fixed and random effects in multiple linear regressions and assessed general model parameters (AIC, log-likelihood, adjusted $R^2$, and the p-value of the likelihood ratio) and searched whether the slopes or intercepts improved. We accounted for collinearity of antibodies against SARS-CoV-2, thus, intrinsic correlation, by using the first PC derived from linear combinations of S, RBD, and NC p (EC50) values (PC1-SARS-CoV-2-IgG). While β2 IgM aPL levels were found to be correlated with the strength of the anti-SARS-CoV-2 antibody response, none of the other features showed significant predictive power. Conversely, PT IgM aPL were best predicted by the strength of the antibody response against SARS-CoV-2 (PC1-SARS-CoV-2-IgG) in combination with sex, as well as disease severity, in patients who contracted SARS-CoV-2. While male patients had a higher average PC1-SARS-CoV-2-IgG, female patients had a propensity to have higher PT IgM aPL values at lower anti-SARS-CoV-2 antibody titres. The feature 'sex', thus, was modelled as a random intercept on PT IgM aPL. These findings were corroborated after exclusion of previously infected convalescent individuals, sampled at later timepoints. Yet, within the group of acutely infected, disease severity was not a relevant additional feature in our LMM.

The detection of antibodies against negatively charged phospholipids and plasma proteins other than CL and β2 may have diagnostic and/or therapeutic consequences. Such aPL e.g., against members of the coagulation cascade like PT or against the PS/PT complex have been described in patients with unprovoked venous thromboembolism [66] seronegative APS [67], and SLE [68]. Although aPL against PT and PS/PT significantly correlate with each other, conformational changes after PT binding to PS may expose different epitopes for aPL binding [68]. Both aPL require different reaction environments for their specific detection [18]. Still, the pathogenicity of these non-criteria (mainly lipid-reactive) aPL associated with SARS-CoV-2 infections is a matter of debate. Recently, a report suggested an association of high-titer aPL (mainly PS/PT) with increased NETosis and more severe respiratory disease in COVID-19 patients [15]. In addition, purified IgG from these patients further led to immune dysregulation and thrombosis in mice [15]. Thus, non-criteria aPL may have pro-thrombotic potential in humans, even if only transiently present.

Molecular mimicry between a pathogenic entity (e.g. virus, bacteria, fungi) and cofactors such as β2 is one of the mechanisms suggested to cause aPL [69]. However, our analyses revealed a rather limited similarity between SARS-CoV-2 Spike protein and PT, and thus suggest a more complex mechanism by which aPT antibodies are elicited. The immunological cascade triggered upon infection with SARS-CoV-2 results in a broad autoantibody response [1,2,4], by a detailed mechanism that yet needs to be established. For the generation of aPT specifically, other proteins involved in the processing, cellular entry and infectivity of SARS-CoV-2 could be of relevance. For instance, one may hypothesise that cross-reactivity between PT and TMPRSS2, a transmembrane serine protease (TTSP) essential for SARS-CoV-2 entry into host cells [70,71] could result in the induction of such antibodies. Recent work further showed that coagulation factors like thrombin cleave SARS-CoV-2 Spike protein and enhance virus entry in cells [72]. However, the impact of aPT on these actions is not known and would need further investigation.

In conclusion, we have conducted an exploratory associative study, with a rigorous follow-up using LMM, to account for variation in all parameters. Our study further emphasizes a

potentially pathogenic role of PT IgM aPL in SARS-CoV-2 infected individuals. We find that its levels significantly correlate with the anti-SARS-CoV-2 antibody response elicited upon infection and is additionally influenced by disease severity and sex. Further studies (possibly multicentric) are needed to assess whether only a specific subset of patients (e.g., genetically defined) would be at risk for developing specific aPL and whether they are linked with thrombotic events.

## Supporting information

**S1 Fig. Demographic features of non-infected controls and SARS-CoV-2 infected individuals. A**. Age distribution of entire cohort for non-infected controls (orange) and SARS-CoV-2 infected individuals (blue). The distribution of the controls indicates a generally younger age versus the SARS-CoV-2 infected individuals. **B**. Sex distribution of entire cohort for non-infected controls (orange) and SARS-CoV-2 infected individuals (blue). The distribution in both cohorts was slightly skewed towards males versus females. **C**. Female (violet) and male (light orange) representation (in %) within each severity score. The distributions are slightly skewed towards more male than female individuals in all severity scores, except for score 2. **D**. Representation of DPO related to severity score. The convalescent individuals included in this study and sampled at later DPO all had a severity score of 1 while the acutely infected hospital patients had severity scores between 2–4.
(TIF)

**S2 Fig. LIA comparison of serum and plasma samples and UMAP representation of aPL profiles. A**. Comparison of patient-matched serum and plasma samples with LIA technology using linear regression. Pearson correlation coefficient is 0.9974 (95% confidence intervals: 09611–1.034) and $R^2$ of 0.9813. **B**. Same as above, visualised as Bland-Altman plot. Bland-Altman analysis indicated a bias of -2.717 (95% confidence interval: -17.53–12.1). Measurements (for A and B) were done as unicates for the entire LIA panel. **C**. and **D**. UMAP of IgM aPL do not reveal any clear cluster between non-infected (black) and SARS-CoV-2 infected (turquoise) individuals, both for IgM (A) as well as for IgG (B) aPL.
(TIF)

**S3 Fig. TRABI comparison of serum and plasma and UMAP representation of SARS-CoV-2 IgG profiles. A**. Binding curves of patient-matched serum and plasma samples. Three SARS-CoV-2 proteins were used to conduct the tripartite-autoimmune blood immunoassay technology: the SARS-CoV-2 spike ectodomain (S), its receptor binding domain (RBD), and the nucleocapsid protein (NC). All samples were tested as technical duplicates and are shown as mean with standard deviation. **B**. Based on the binding curves, the concentration at half-maximum binding of the sigmoidal curve, or the p(EC50) value, was calculated. p(EC50) values for S, RBD, and NC for all patients are compared in a scatter plot with serum-based p(EC50) values on the x-axis and plasma-based p(EC50) values on the y-axis. The Pearson correlation coefficient was 0.9942 (95% confidence interval: 0.9865–0.9975) and $R^2$ was 0.9885. Dotted line represents the 95% confidence interval of the linear regression. **C**. Visualisation of the same data as shown in (B) using a Bland-Altman plot. Bias was calculated to be 0.01976 with a standard deviation of 0.1361. Dotted lines represent the confidence intervals (-0.2471 to 0.2866). **D**. UMAP representation of anti-SARS-CoV-2 protein antibodies displays clear clusters, with non-infected controls (black) and non-IgG-reactive SARS-CoV-2 infected individuals (turquoise) clustering separately from IgG-reactive SARS-CoV-2 infected individuals.
(TIF)

**S4 Fig. Age, severity, and DPO in relation to PT and β2 IgM levels. A**. Severity score as a function of age, with colour-coded PT (left) and β2 (right) IgM levels. When ignoring the interdependency of features, data suggests that PT and β2 IgM levels are modulated by severity but not by age. **B**. PT and β2 IgM levels as a function of DPO. PT IgM aPL levels show a non-homogenous distribution along DPO, with a rise between DPO 10–30 and a subsequent drop. β2 IgM levels, on the other hand, seem stable over time. To summarize the data, aPL levels were binned in 15 day windows. The LMM used in this study takes care of the interdependency of parameters and accounts for redundancy, in the present case e.g., between severity score and DPO and their effect on PT or β2 IgM titres. Values are shown as single dots, with boxplot (median and interquartile range) in (A) and as mean ± standard deviation in (B).
(TIF)

**S1 Table. Equation and performance characteristics of multiple linear fixed-effect and mixed-effect models using data from non-infected controls and SARS-CoV-2 infected individuals.** The best model is shown in bold letters (AIC, Akaike information criterion; β2, β2-glycoprotein I; DPO, day post onset; PC, principal component).
(DOCX)

**S2 Table. Equation and performance characteristics of multiple linear fixed-effect and mixed-effect models using data from non-infected controls and SARS-CoV-2 infected individuals, without including data from convalescent individuals.** The best model is shown in bold letters (AIC, Akaike information criterion; DPO, day post onset; PC, principal component; PT, prothrombin).
(DOCX)

**S3 Table. Equation and performance characteristics of multiple linear fixed-effect and mixed-effect models using data from SARS-CoV-2 infected individuals without the non-infected controls.** The best model is shown in bold letters (AIC, Akaike information criterion; DPO, day post onset; PC, principal component; PT, prothrombin).
(DOCX)

**S1 Data. Raw data underlying this study is provided as an archive file.** The archive contains the data for the comparison between serum and plasma for TRABI and LIA as well as the experimental data (LIA and TRABI), including the most important epidemiological features of patients.
(ZIP)

## Acknowledgments

We thank Dr. Sumana Srivatsa (former ETH Zurich, now Stanford University, California) for valuable discussions on data models at the initial stage of data exploration, Anne Hollerbach (University Medical Center of the Johannes Gutenberg University, Mainz, Germany) for providing matched plasma and serum samples of aPL-positive patients, and Prof. Dr. Andreas Hierlemann (BEL, D-BSSE, ETH Zurich) for support.

## Author Contributions

**Conceptualization:** Marc Emmenegger, Katrin B. M. Frauenknecht.

**Data curation:** Marc Emmenegger, Sreedhar Saseendran Kumar, Vishalini Emmenegger, Dirk Roggenbuck, Katrin B. M. Frauenknecht.

**Formal analysis:** Marc Emmenegger, Sreedhar Saseendran Kumar, Vishalini Emmenegger, Dirk Roggenbuck, Katrin B. M. Frauenknecht.

**Funding acquisition:** Marc Emmenegger, Adriano Aguzzi.

**Investigation:** Marc Emmenegger, Tomas Malinauskas, Laura Rose, Dirk Roggenbuck, Katrin B. M. Frauenknecht.

**Methodology:** Marc Emmenegger, Adriano Aguzzi, Dirk Roggenbuck.

**Project administration:** Marc Emmenegger, Katrin B. M. Frauenknecht.

**Resources:** Peter Schierack, Martin F. Sprinzl, Clemens J. Sommer, Karl J. Lackner, Adriano Aguzzi, Dirk Roggenbuck, Katrin B. M. Frauenknecht.

**Software:** Sreedhar Saseendran Kumar, Vishalini Emmenegger, Dirk Roggenbuck.

**Supervision:** Adriano Aguzzi, Dirk Roggenbuck.

**Validation:** Marc Emmenegger, Sreedhar Saseendran Kumar, Vishalini Emmenegger.

**Visualization:** Marc Emmenegger, Sreedhar Saseendran Kumar, Vishalini Emmenegger, Tomas Malinauskas.

**Writing – original draft:** Marc Emmenegger, Sreedhar Saseendran Kumar, Vishalini Emmenegger, Dirk Roggenbuck, Katrin B. M. Frauenknecht.

**Writing – review & editing:** Marc Emmenegger, Sreedhar Saseendran Kumar, Vishalini Emmenegger, Tomas Malinauskas, Thomas Buettner, Laura Rose, Peter Schierack, Martin F. Sprinzl, Clemens J. Sommer, Karl J. Lackner, Adriano Aguzzi, Dirk Roggenbuck, Katrin B. M. Frauenknecht.

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
