## [Decision Letter · Decision Letter 0]

25 Oct 2021

Dear Dr. Frauenknecht,

Thank you very much for submitting your manuscript "Anti-prothrombin autoantibodies enriched after infection with SARS-CoV-2 and influenced by strength of antibody response against SARS-CoV-2 proteins" for consideration at PLOS Pathogens. As with all papers reviewed by the journal, your manuscript was reviewed by members of the editorial board and by several independent reviewers. The reviewers appreciated the attention to an important topic. Based on the reviews, we are likely to accept this manuscript for publication, providing that you modify the manuscript according to the review recommendations.

Reviewer #2 raises some concerns that will need to be addressed before further consideration. In particular, it is important to discuss the validity of the data incorporated in the statistical model as well as the time-dependence of the observations made regarding aPL IgG and SARS-CoV-2-specific antibody responses.

Sincerely,

Ali Ellebedy

Guest Editor

PLOS Pathogens

Carolina Lopez

Section Editor

PLOS Pathogens

Kasturi Haldar

Editor-in-Chief

PLOS Pathogens

orcid.org/0000-0001-5065-158X

Michael Malim

Editor-in-Chief

PLOS Pathogens

orcid.org/0000-0002-7699-2064

Reviewer #2 raises some concerns that will need to be addressed before further consideration. In particular, it is important to discuss the validity of the data incorporated in the statistical model as well as the time-dependence of the observations made regarding aPL IgG and SARS-CoV-2-specific antibody responses.

Reviewer Comments (if any, and for reference):

Reviewer's Responses to Questions

**Part I - Summary**

Reviewer #1: I'd like to thank the authors for addressing my concerns with added data and text. I think the manuscript has significantly improved. I do not have any further comments.

Reviewer #2: Summary

This study investigated the presence of antiphospholipid antibodies (aPL) in different stages of COVID-19 in different patients at different stages of disease or recovery. The aetiology of hypercoagulability in COVID-19 is a pressing question given the severe morbidity and mortality that arises and the need to find appropriate treatments. Research in the field is important.

The study has already been reviewed once, and the authors have responded, where possible, to some of the reviewers’ comments. This is evidenced by demonstration of the interchangeable use of serum or plasma in serology assays, and the sequence analysis of SARS-CoV-2 spike and human prothrombin. However, the original limitations in the study design and the interpretation of data that are associative rather than tractable remain weaknesses in the manuscript.

**Part II – Major Issues: Key Experiments Required for Acceptance**

Reviewer #1: (No Response)

Reviewer #2: As pointed out by reviewer number 2, the cohort is highly heterogenous. Combining the analysis of sera/plasma from individuals who have survived COVID-19 and who are in the convalescent phase (and may therefore bear features of survival) with those suffering from acute illness is illogical. Serological diagnosis is time dependent and this forms the basis of acute and convalescent testing in infectious disease. Furthermore, induction of IgM and IgG is time sensitive. The time dependency of humoral responses is observable in Figure 3A where OD of PT IgM, PC1 SARS-CoV-2 IgG and B2Igm are generally lower in those further from disease onset. The lack of a relationship between aPL IgG and SARS-CoV-2 -specific antibody combined suggests a transient acute response. These features of serological evolution have not been appropriately considered in the study design.

Whilst a statistical model has been used to try to unpick heterogeneity, concise clear case definition, alongside timing of diagnosis and PCR testing prior to data analysis would be preferable. Table 3 has not been divided into the two groups amongst data from those defined as SAR-CoV-2 infected. Reviewer #2 point 6 that there are concerns about the data put into the modelling, which brings into question the validity of the conclusions remains unresolved.

Given the possibility that aPL are associated with the hypercoagulable state, case definition would be strengthened by an understanding of those where there was demonstrable clinical evidence of this e.g. pulmonary embolism.

The data interpretation does not take account of literature demonstrating a link between induction of aPL and several different infections, including where a vasculitis or endothelial injury occurs. For example, infection with Treponema pallidum is associated with the induction of anticardiolipin antibodies which are used as a surrogate measure of infection, disease activity and re-infection. However, these patients do not typically enter a hypercoagulable state, possibly due to differences in binding specificity of antibodies in antiphospholipid syndrome compared with syphilis (Pierangeli S. et al 1994). It is possible for aPL antibodies to bind with no functional significance for clotting and causality cannot be assumed. The line immunoassay does not measure antibody function in this regard. Can further studies be done to assess this?

Figure 3A suggests a link with female sex and PT IgM. COVID-19 is more severe in men. This apparent paradox is not discussed. It is also surprising that no effect of age was observed given the striking linear association of age with mortality from COVID-19.

Reviewer #3 comment that this study may explain the development of thrombophilia after SARS-CoV-2 vaccination is not valid. There is no vaccinated cohort in this dataset in which to study this phenomenon.

Given the literature indicating aPL in multiple unrelated infections, it is unsurprising that there is no homology between anti-SARS-CoV-2 antibodies and aPL. Although molecular mimicry is a possibility, it is also possible that release of host phospholipids through cellular damage could induce aPL. However, it is helpful that the authors have done some work to investigate this in Fig 4.

Several other studies have investigated aPL in COVID-19 and these have been published with conflicting data and interpretation. The findings from this study should be placed in the context of what has already been published.

**Part III – Minor Issues: Editorial and Data Presentation Modifications**

Reviewer #1: (No Response)

Reviewer #2: Part III – Minor Issues: Editorial and Data Presentation Modifications

Please check the manuscript for wording and grammatical errors, particularly in the abstract.

PLOS authors have the option to publish the peer review history of their article (what does this mean?). If published, this will include your full peer review and any attached files.

Reviewer #1: No

Reviewer #2: **Yes: **Katrina Mary Pollock

Figure Files:

Data Requirements:

Reproducibility:

References:

---

## [Editor Report · Decision Letter 1]

15 Nov 2021

Dear Dr. Frauenknecht,

We are pleased to inform you that your manuscript 'Anti-prothrombin autoantibodies enriched after infection with SARS-CoV-2 and influenced by strength of antibody response against SARS-CoV-2 proteins' has been provisionally accepted for publication in PLOS Pathogens.

Best regards,

Ali Ellebedy

Guest Editor

PLOS Pathogens

Carolina Lopez

Section Editor

PLOS Pathogens

Kasturi Haldar

Editor-in-Chief

PLOS Pathogens

orcid.org/0000-0001-5065-158X

Michael Malim

Editor-in-Chief

PLOS Pathogens

orcid.org/0000-0002-7699-2064

Thanks for addressing Reviewers' concerns/comments
---

## [Editor Report · Acceptance letter]

26 Nov 2021

Dear Dr. Frauenknecht,

We are delighted to inform you that your manuscript, "Anti-prothrombin autoantibodies enriched after infection with SARS-CoV-2 and influenced by strength of antibody response against SARS-CoV-2 proteins," has been formally accepted for publication in PLOS Pathogens.

Best regards,

Kasturi Haldar

Editor-in-Chief

PLOS Pathogens

orcid.org/0000-0001-5065-158X

Michael Malim

Editor-in-Chief

PLOS Pathogens

orcid.org/0000-0002-7699-2064